# A Graph Neural Network Assisted Monte Carlo Tree Search Approach to Traveling Salesman Problem

## Abstract

We present a graph neural network assisted Monte Carlo Tree Search approach for the classical traveling salesman problem (TSP). We adopt a greedy algorithm framework to construct the optimal solution to TSP by adding the nodes successively. A graph neural network (GNN) is trained to capture the local and global graph structure and give the prior probability of selecting each vertex every step. The prior probability provides a heuristic for MCTS, and the MCTS output is an improved probability for selecting the successive vertex, as it is the feedback information by fusing the prior with the scouting procedure. Experimental results on TSP up to 100 nodes demonstrate that the proposed method obtains shorter tours than other learning-based methods.

## 1 Introduction

Traveling Salesman Problem (TSP) is a classical combinatorial optimization problem and has many practical applications in real life, such as planning, manufacturing, genetics (Applegate et al., 2006b). The goal of TSP is to find the shortest route that visits each city once and ends in the origin city, which is well-known as an NP-hard problem (Papadimitriou, 1977). In the literature, approximation algorithms were proposed to solve TSP (Lawler et al., 1986; Goodrich & Tamassia, 2015). In particular, many heuristic search algorithms were made to find a satisfactory solution within a reasonable time. However, the performance of heuristic algorithms depends on handcrafted heuristics to guide the search procedure to find competitive tours efficiently, and the design of heuristics usually requires substantial expertise of the problem (Johnson & McGeoch, 1997; Dorigo & Gambardella, 1997).

Recent advances in deep learning provide a powerful way of learning effective representations from data, leading to breakthroughs in many fields such as speech recognition (Lecun et al., 2015). Efforts of the deep learning approach to tackling TSP has been made under the supervised learning and reinforcement learning frameworks. Vinyals *et al.* (Vinyals et al., 2015) introduced a pointer network based on the Recurrent Neural Network (RNN) to model the stochastic policy that assigns high probabilities to short tours given an input set of coordinates of vertices. Dai *et al.* (Dai et al., 2017) tackled the difficulty of designing heuristics by Deep Q-Network (DQN) based on structure2vec (Dai et al., 2016b), and a TSP solution was constructed incrementally by the learned greedy policy. Most recently, Kool *et al.* (Kool et al., 2019) used Transformer-Pointer Network (Vaswani et al., 2017) to learn heuristics efficiently and got close to the optimal TSP solution for up to 100 vertices. These efforts made it possible to solve TSP by an end-to-end heuristic algorithm without special expert skills and complicated feature design.

In this paper, we present a new approach to solving TSP. Our approach combines the deep neural network with the Monte Carlo Tree Search (MCTS), so that takes advantage of the powerful feature representation and scouting exploration. A graph neural network (GNN) is trained to capture the local and global graph structure and predict the prior probability, for each vertex, of whether this vertex belongs to the partial tour. Besides node features, we integrate edge information into each update-layer in order to extract features efficiently from the problem whose solution relies on the edge weight.

Similar to above-learned heuristic approaches, we could greedily select the vertex according to the biggest prior probability and yet the algorithm may fall into the local optimum because the algorithm has only one shot to compute the optimal tour and never goes back and reverses the decision. To overcome this problem, we introduce a graph neural network assisted Monte Carlo Tree Search (GNN-MCTS) to make the decision more reliable by a large number of scouting simulations. The trained GNN is used to guide the MCTS procedure that effectively reduces the complexity of the search space and MCTS provides a more reliable policy to avoid stuck in a local optimum. Experimental results on TSP up to 100 vertices demonstrate that the proposed method obtains shorter tours than other learning-based methods.

The remainder of the paper is organized as follows: After reviewing related work in Section 2, we briefly give a preliminary introduction to TSP in Section 3. Our approach is formulated in Section 4. Experimental results are given in Section 5, followed by the conclusion in Section 6.

## 2 RELATED WORK

The TSP is a well studied combinatorial optimization problem, and many learning-based algorithms have been proposed. In 1985, Hopfield *et al.* proposed a neural network to solve the TSP (Hopfield & Tank, 1985). This is the first time that researchers attempted to use neural networks to solve combinatorial optimization problems. Since the impressive results produced by this approach, many researchers have made efforts on improving the performance (Bout & Miller, 1988; Brandt et al., 1988). Many shallow network architectures were also proposed to solve the combinatorial optimization problem (Favata & Walker, 1991; Fort, 1988; Angniol et al., 1988; Kohonen, 1982). Recent years, deep neural networks have been adopted to solve the TSP and many works have achieved remarkable results. We summarize the existing learning-base methods from the following aspects.

### ENCODER AND DECODER

Vinylas *et al.* (Vinyals et al., 2015) proposed a neural architecture called Pointer Net (Ptr-Net) to learn the conditional probability of a tour using a mechanism of the neural attention. Instead of using attention to blend hidden units of an encoder to a context vector, they used attention as pointers to the input vertices. The parameters of the model are learned by maximizing the conditional probabilities for the training examples in a supervised way. Upon test time, they used a beam search procedure to find the best possible tour. Two flaws exist in the method. First, Ptr-Net can only be applied to solve problems of a small scale ($n \leq 50$). Second, the beam search procedure might generate invalid routes.

Bello *et al.* (Bello et al., 2017) proposed a framework to tackle TSP using neural networks and reinforcement learning. Similar to Vinylas *et al.*, they employed the approach of Ptr-Net as a policy model to learn a stochastic policy over tours. Furthermore, they masked the visited vertices to avoid deriving invalid routes and added a glimpse which aggregates different parts of the input sequence to improve the performance. Instead of training the model in a supervised way, they introduced an Actor-Critic algorithm to learn the parameters of the model and empirically demonstrated that the generalization is better compared to optimizing a supervised mapping of labeled data. The algorithm significantly outperformed the supervised learning approach (Vinyals et al., 2015) with up to 100 vertices.

Kool *et al.* (Kool et al., 2019) introduced an efficient model and training method for TSP and other routing problems. Compared to (Bello et al., 2017), they removed the influence on the input order of the vertices by replacing recurrence (LSTMs) with attention layers. The model can include valuable information about the vertices by multi-head attention mechanism which plays an important role in the setting where decisions relate directly to the vertices in a graph. Similar to (Bello et al., 2017), they applied a reinforcement learning method to train the model. Instead of learning a value function as a baseline, they introduced a greedy rollout policy to generate baseline and empirically showed that the greedy rollout baseline can improve the quality and convergence speed for the approach. They improved the state-of-art performance among 20, 50, and 100 vertices. Independent of the work of Kool *et al.*, Deudon *et al.* (Deudon et al., 2018) also proposed a framework which uses attention layers and reinforcement learning algorithm (Actor-Critic) to learn a stochastic policy.

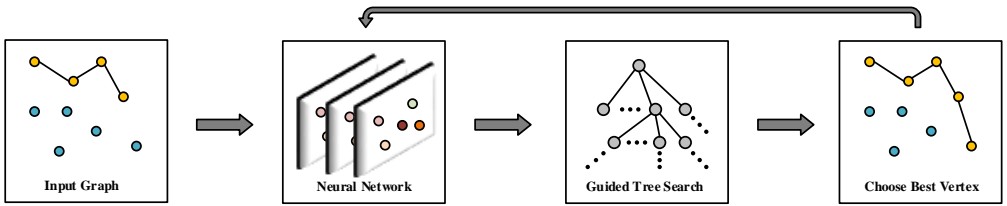

Figure 1: Approach overview. First, the graph is fed into the graph neural network, which captures global and local graph structure and generates a prior probability that indicates how likely each vertex is in the tour sequence. Then, with the help of the graph neural network, a developed MCTS outputs an improved probability by scouting simulations. Lastly, we visit the best vertex among unvisited vertices according to the improved probability. The above process will loop until all vertices are visited.

They combined the machine learning methods with an existing heuristic algorithm, i.e., 2-opt to enhance the performance of the framework.

GRAPH EMBEDDING

Dai *et al.* (Dai et al., 2017) proposed a framework, which combines reinforcement learning with graph embedding neural network, to construct solutions incrementally for TSP and other combinatorial optimization problems. Instead of using a separate encoder and decoder, they introduced a graph embedding network based on the structure2vec (Dai et al., 2016a) to capture the current state of the solution and the structure of a graph. Furthermore, they used Q-learning parameterized by the graph embedding network to learn a greedy policy that outputs which vertex being inserted into the partial tour. They adopt the farthest strategy (Rosenkrantz et al., 2013) to get the best insertion position of the partial tour.

Nowak *et al.* (Nowak et al., 2017) propose a supervised manner to directly output a tour as an adjacency matrix based on a Graph Neural Network and then convert the matrix into a feasible solution by beam search . The author only reports an optimality gap of 2.7% for $n = 20$ and slightly worse than the auto-regressive data-driven model (Vinyals et al., 2015).

The performance of the above-mentioned methods was suffered due to the greedy policy which selects the vertex according to the biggest prior probability or the value. In this paper, we introduce a new Monte Carlo Tree Search-based algorithm to overcome this problem.

## 3  PRELIMINARIES

TRAVELING SALESMAN PROBLEM

Let $G(V, E, w)$ denotes a weighted graph, where $V$ is the set of vertices, $E$ the set of edges and $w : E \rightarrow R^+$ the edge weight function, i.e., $w(u, v)$ is the weight of edge $(u, v) \in E$. We use $S = \{v_1, v_2, ..., v_i\}$ to represent an ordered tour sequence that starts with $v_1$ and ends with $v_i$, and $\bar{S} = V \setminus S$ the set of candidate vertices for addition, condition on $S$. The target of TSP is to find a tour sequence with the lowest cost, i.e., $c(G, S) = \sum_{i=1}^{|S|-1} w(S(i), S(i + 1)) + w(S(|S|), S(1))$ when $|S| = |V|$.

## 4  PROPOSED APPROACH

For a graph, our goal is to construct a tour solution by adding vertices successively. A natural approach is to train a deep neural network of some form to decide which vertex being added to the partial tour at each step. That is, a neural network $f$ would take the graph $G$ and the partial tour sequence $S$ as input, and the output $f(G|S)$ would be a prior probability that indices how likely each vertex to be selected. Intuitively, we can use the prior probability in a greedy way, i.e., selecting vertex with the biggest probability, to generate the tour sequence incrementally. However,

deriving tours in this way might fall into the local optimum because the algorithm has only one shot to compute the optimal tour and never goes back and reverses the decision. To overcome this problem, we enhance the policy-decisions by MCTS assisted with the deep neural network.

We begin in Section 4.1 by introducing how to transform TSP into a Markov Decision Process (MDP). Then in Section 4.2, we describe the GNN architecture for parameterizing $f(G|S)$. Finally, Section 4.3 describes GNN-MCTS for combinatorial optimization problems, especially the TSP. The overall approach is illustrated in Figure 1.

## 4.1 Traveling Salesman Problem as Markov Decision Process

We present TSP as a MDP as follows,

- **States**: a state $s$ is an ordered sequence of visited vertices on a graph $G$ and the terminal state is that all vertices have been visited.

- **Transition**: transition is deterministic in the TSP, and corresponds to adding one vertex $v \in \bar{S}$ to $S$.

- **Actions**: an action $a$ is selecting a vertex of $G$ from the vertices candidate set $\bar{S}$.

- **Rewards**: the reward function $r(s, a)$ at state $s$ is defined as the change of cost after taking action $a$ and transitioning to a new state $s'$, i.e., $r(s, a) = -w(v_m, v_n)$, where $v_m$ and $v_n$ are the last vertex in partial tour sequence $S$ and $S'$ respectively.

- **Policy**: based on the improved probability $\hat{P}$ generated by the GNN-MCTS, a deterministic greedy policy $\pi(v|S) := \arg\max_{v' \in \bar{S}} \hat{P}(S, v')$ is used.

## 4.2 Deep Neural Network Architecture

To compute a good policy, information about the global structure of the graph and the current constructed tour sequence $S = \{v_1, ..., v_i\}$ is required. We tag the nodes which have been visited as $x_v = 1$. Intuitively, $f(G|S)$ should summarize the state of such a "tagged" graph and generate the prior probability that indicates how likely each vertex is to belong to $S$. It is challenging to design a neural network $f(G|S)$ to capture local and global graph structure. In order to represent such a complicated context, we propose a new deep learning architecture based on graph neural networks (GNN) to parameterize $f(G|S)$.

### Graph Neural Networks

Similar to the basic GNN, we design the neural network $f(G|S; \Theta)$ to compute a $l$-dimensional feature $H_v$ for each vertex of a "tagged" graph. We use $H_v^t$ to denote the real-valued feature vector associated with $v$ after the computation by the layer $t$. A GNN model consists of a stack of $T$ neural network layers, where each layer aggregates local neighborhood information, i.e., features of neighbors around each node, and then passes this aggregated information on to the next layer. Specifically, the basic GNN model (Hamilton et al., 2017) can be implemented as follows. In each layer $t \in [0, T]$, a new feature is computed as:

$$H_v^{t+1} = \sigma \left( H_v^t W_1^t + \sum_{u \in \mathcal{N}(v)} H_u^t W_2^t \right) \tag{1}$$

where $\mathcal{N}(v)$ is the set of neighbors of vertex $v$, $W_1^t$ and $W_2^t$ are parameter matrices for the layer $t$, and $\sigma$ denotes a component-wise non-linear function, e.g., a sigmoid or a ReLU. For $t = 0$, $H_v^0$ denotes the feature initialization at the input layer.

The above GNN architecture has been demonstrated to perform well on combinatorial optimizations problems such as Maximal Independent Set (MIS), Minimum Vector Cover (MVC), etc. (Li et al., 2018). As observed from the Equation 1 , the edge information is not taken into account for MIS, MVC, but, for TSP, edge information cannot be ignored, because the object of TSP is computed based on the edge cost, i.e., the distance between the two vertices. We integrate edge information

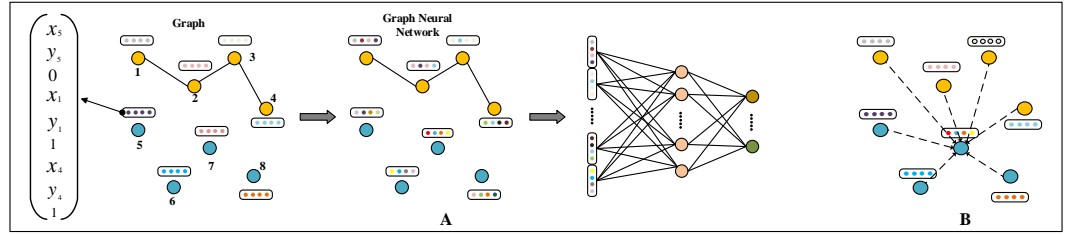

Figure 2: Neural network architecture. The architecture on the left (A) is used to compute the prior probability map that indicates how likely each vertex is in the tour sequence. Firstly, the "tagged" graph is fed into the GNN to generate new feature expressions for each vertex. Then all new node feature is concentrated into a long vector that denotes the context of the "tagged" graph. Lastly, the vector is fed into a multilayer perceptron to output the prior probability. The picture on the right (B) depicts the mechanism of computing a new feature of the vertex in one update-layer.

into the new node feature $H$ as follows,

$$H_v^{t+1} = \sigma \left( H_v^t W_1^t + \sum_{u \in \mathcal{N}(v)} H_u^t W_2^t + \frac{1}{|\mathcal{N}(v)|} \sum_{u \in \mathcal{N}(v)} e_{v,u} W_3^t \right) \quad (2)$$

where $e(v, u)$ is the distance[1] between two vertices and $W_3^t$ are parameter matrices for the layer $t$.

Dai et al.(Dai et al., 2017) proposed a graph embedding networks (GEN) based on structure2vec to compute new node feature $\mu$ as follows,

$$\mu_v^{t+1} = \text{relu} \left( \theta_1 x_v + \theta_2 \sum_{u \in \mathcal{N}(v)} \mu_u^t + \theta_3 \sum_{u \in \mathcal{N}(v)} \text{relu}(\theta_4 w(v, u)) \right) \quad (3)$$

where $\theta_1 \in \mathbb{R}^l$, $\theta_2, \theta_3 \in \mathbb{R}^{l \times l}$ and $\theta_4 \in \mathbb{R}^l$ are model parameters.

Compared with GEN, the key improvements are: 1) Our GNN replaces $x_v$ in Equation 3 with $H_v$ so that the our GNN could integrate the latest feature of the node itself directly in each update procedure. 2) One can regard each update process in the GEN as one update layer of the our GNN, i.e., each calculation is equivalent to going one layer forward, and counting T times is the T layers. Parameters of each layer in our GNN are independent, while parameters are shared between different update processes in GEN which limits the ability of the neural network. 3) Instead of aggregating edge weight by "sum" operation, we use "average" operation to balance the weight of node and edge feature. Experimental results show that the above improvements enhance the performance of the neural network.

We initialize the node feature $H^0$ as follows. Each vertex has a feature tag which is a 3-dimensional vector. The first element is binary and equal to 1 if the partial tour sequence $S$ contains the vertex. The second and third elements of the feature tag are the coordinates of the vertex. When a partial tour has been constructed, it can not be changed, and the remaining problem is to find a path from the last vertex, through all unvisited vertices, to the first vertex. To know the first and the last vertex in partial tour sequence $S$, besides basic feature tags described above, we extend the node feature $H^0$ by adding feature tags of the first and last vertex in partial tour sequence $S$ (see in Figure 2).

PARAMETERIZING $f(G|S; \Theta)$

Once feature for each vertex is computed after $T$ iterations, and we use the new feature of vertices to define the $f(G|S; \Theta)$, which outputs the prior probability indicating how likely each vertex is to belong to partial tour sequence $S$. More specifically, we fuse all vertex feature $H_v^T$ as the current

---

[1]Euclidean distance: given two points $(x_1, y_1)$ and $(y_1, y_2)$ in two-dimensional plane, $D = \sqrt{(x_2 - x_1)^2 + (y_2 - y_1)^2}$

state representation of the graph and parameterize $f(G|S; \Theta)$ as follows ,

$$f(G|S; \Theta) = \text{softmax}(sum(H_1^T), ..., sum(H_n^T)) \tag{4}$$

where $sum$ denotes summation operator.

During training, we minimize the cross-entropy loss for each training sample $(G_i, S_i)$:

$$\ell(S_i, f(G_i|S_i; \Theta)) = -\sum_{j=1}^{N} y_j \log f(G_i|S_i(1:j-1); \Theta) \tag{5}$$

where $S_i$ is a tour sequence which is a permutation of the vertices over graph $G_i$ and $y_j$ is a one-hot vector whose length is $N$ and $S(j)$-th position is 1.

The architecture of the deep neural networks is illustrated in Figure 2.

### 4.3 Graph neural network assisted Monte Carlo tree search

Similar to the implementation in (Silver et al., 2016), the GNN-MCTS uses deep neural networks as a guide. Each node $s$ in the search tree contains edges $(s, a)$ for all legal actions $a \in A(s)$. Each edge stores a set of statistics,

$$\{N(s, a), Q(s, a), P(s, a)\}$$

where $N(s, a)$ is the visit count, $Q(s, a)$ is the action value and $P(s, a)$ is the prior probability of selecting that edge.

To be mentioned, three biggest differences between GNN-MCTS and AlphaGo are:

- When playing the game of Go, the branch with a high average rate of winning indicates that the route is strong. While TSP is interested in finding the extreme, the average value makes no sense if several suboptimal routes surround the extreme route. Instead of recording the average action value, we propose to track the best action value found under each node's subtree for determining its exploitation value.

- In the game of Go, it is common to use $\{0, 0.5, 1\}$ to denote the result of a game composed of $loss$, $draw$, and $win$. Not only is this convenient, but it also meets the requirements of UCT (Kocsis & Szepesvári, 2006) for rewards to lie in the $[0, 1]$ range. In TSP, an arbitrary tour length can be achieve that does not fall into the predefined interval. One can solve this issue by adjusting the parameter $c_{puct}$ of UCT in such a way that it is feasible for a specified interval. It requires substantial trial-and-error on adjusting $c_{puct}$ due to the change in the number of cities. Instead, we address this problem by normalizing the action value of each node $n$ whose parent is node $p$ to $[0, 1]$ as follows,

$$Q_n = \frac{Q_n - w_p}{b_p - w_p} \tag{6}$$

where $b_p$ and $w_p$ are, respectively, the best (maximum) and the worst (minimum) action value under $p$, and $Q_n$ is the action value of $n$. The best action value under $p$ is normalized to the value of 1, the worst action value is normalized to 0, and all other results are normalized to $[0, 1]$.

- AlphaGo used a learned value function (critic) $v(s, \theta)$ to estimate the probability of the current player winning from position $s$, where the parameters $\theta$ are learned from the observations $(s, \pi)$. However, getting such algorithms to work is non-trivial. Instead, we design a value function $h(s)$ that combines the GNN and beam search to evaluate the possible tour length from the current state to the end state. Guided by the output of GNN, the value function executes beam search from the state corresponding to the leaf node $l$ until reaching an end state. We compute the value of leaf node $V_l$ according to the partial tour sequence $S$ corresponding to the end state as follows,

$$V_l = -\left(\sum_{i=1}^{|S|-1} w\left(S(i), S(i+1)\right) + w\left(S(|S|), S(1)\right)\right) \tag{7}$$

The value function is described in algorithm 1.

The GNN-MCTS proceeds by iterating over the four phases and then selects a move to play.

---

[2] value = $\prod_{j=1}^{|S|} f(G|S(1:j-1))$, where $S$ is the partial tour sequence corresponding to the state.

---

**Algorithm 1 Value Function**

---

$start$ denotes the state of leaf node $l$

$B$ denotes the beam width

1: Initialize BEAM = $\{start\}$
2: **while** BEAM $\neq \varnothing$ **do**
3:      SET = $\varnothing$
4:      **for** state in BEAM **do**
5:         **for** successor of state **do**
6:           Compute value[2] of the successor
7:           SET = SET $\cup$ { successor }
8:      BEAM = $\varnothing$
9:      **while** SET $\neq \varnothing$ and $B < |$SET$|$ **do**
10:         state = successor in SET with smallest value
11:         SET $\setminus$ { state }
12:      **for** state in SET **do**
13:         **if** state is end **then**
14:           **return** state in SET with biggest value
15:         **else**
16:           BEAM = BEAM $\cup$ { state }

---

***Selection Strategy***. The first in-tree phase of each rollouts begins at the root of node $s_0$ of the search tree and finishes when the rollouts reaches a leaf node $s_l$ at time step $l$. At each of these time steps, $t<l$, we use a variant of PUCT (Rosin, 2011) to balance exploration(i.e., visiting states suggested by the prior policy) and exploitation(i.e., visiting states that have the best value) according to the statistics in the search tree.

$$a_t = \arg\max_a \left(Q(s_t, a) + U(s_t, a)\right) \tag{8}$$

$$U(s, a) = c_{puct} P(s, a) \frac{\sqrt{\sum_b N(s, b)}}{1 + N(s, a)} \tag{9}$$

where $c_{puct}$ is a constant to trading off between exploration and exploitation.

***Expansion Strategy***. When a leaf node $l$ is reached, the corresponding state $s_l$ is evaluated by the deep neural network to obtain the prior probability $p$ of its child nodes. The leaf node is expanded and the statistic of each edge $(s_l, a)$ is initialized to $\{N(s_l, a) = 0, Q(s_l, a) = -\infty^3, P(s_l, a) = p_a\}$.

***Simulation Strategy***. Rather than using a random strategy, we use value function $h(s)$ to evaluate the length of the tour that may be generated from the leaf node $s_l$.

***Back-Propagation Strategy***. For each step $t < L$, the edge statistics are updated in a backward process. The visit counts are increased, $N(s_t, a_t) = N(s_t, a_t) + 1$, and the action value is updated to best value, $Q(s_t, a_t) = \max(Q(s_t, a_t), V_l)$.

***Play***. At the end of several rollouts, we select node with the biggest $\hat{P}(a|s_0) = 1 - \frac{Q(s_0, a)}{\sum_b Q(s_0, b)}$ as the next move $a$ in the root position $s_0$. The search tree will be reused at subsequent time steps: the child node relating to the selected node becomes the new root node, and all the statistics of sub-tree below this child node is retained.

---

[3] In the experiment, we initialize the Q to -5.0, -10.0 and -15.0 respectively for TSP20, TSP50, and TSP100.

Table 1: Our method vs baselines. The gap % is w.r.t. the best value across all methods

| | Random | | | | | | Clustered | | | | | |
|---|---|---|---|---|---|---|---|---|---|---|---|---|
| Method | n=20 | | n=50 | | n=100 | | n=20 | | n=50 | | n=100 | |
| | Obj. | Gap | Obj. | Gap | Obj. | Gap | Obj. | Gap | Obj. | Gap | Obj. | Gap |
| Concorde | 3.92 | 0.0% | 5.68 | 0.0% | 7.73 | 0.0% | 3.30 | 0.0% | 3.38 | 0.0% | 3.39 | 0.0% |
| Gurobi | 3.92 | 0.0% | 5.68 | 0.0% | 7.73 | 0.0% | 3.30 | 0.0% | 3.38 | 0.0% | 3.39 | 0.0% |
| Nearest Neighbor | 4.57 | 16.50% | 7.02 | 23.44% | 9.63 | 24.58% | 3.95 | 19.48% | 4.18 | 23.76% | 4.23 | 25.02% |
| Nearest Insertion | 4.40 | 12.24% | 6.77 | 19.08% | 9.48 | 22.64% | 3.66 | 10.62% | 3.97 | 17.36% | 4.08 | 20.46% |
| Random Insertion | 4.08 | 4.12% | 6.09 | 7.12% | 8.45 | 9.24% | 3.46 | 4.67% | 3.65 | 7.93% | 3.72 | 9.70% |
| Farthest Insertion | 4.03 | 2.73% | 5.98 | 5.29% | 8.33 | 7.78% | 3.40 | 2.87% | 3.58 | 5.91% | 3.64 | 7.55% |
| Vinyals et al. (gr.) | 3.97 | 1.30% | 6.41 | 18.58% | - | | - | | - | | - | |
| Bello et al. (gr.) | 3.99 | 1.83% | 5.95 | 4.75% | 8.26 | 6.82% | - | | | | - | |
| Kool et al. (gr.) | 3.93 | 0.36% | 5.78 | 1.75% | 8.08 | 4.57% | - | | - | | - | |
| Dai et al. | 4.03 | 2.76% | 5.98 | 5.26% | 8.33 | 7.76% | 3.37 | 2.07% | 3.58 | 6.01% | 3.66 | 8.17% |
| GNN-MCTS | **3.92** | **0.03%** | **5.70** | **0.32%** | **7.85** | **1.53%** | **3.30** | **0.00%** | **3.39** | **0.34%** | **3.44** | **1.61%** |
| MCTS | 6.37 | 62.45% | 18.43 | 224.28% | 40.27 | 420.83% | 4.98 | 50.69% | 9.26 | 173.99% | 14.00 | 313.33% |
| GEN-MCTS | 3.92 | 0.12% | 5.75 | 1.31% | 8.08 | 4.53% | 3.31 | 0.21% | 3.42 | 1.04% | 3.55 | 4.72% |
| GNN-MCTS-t | 4.00 | 2.21% | 5.98 | 5.26% | 8.61 | 11.34% | 3.37 | 2.19% | 3.57 | 5.61% | 3.73 | 10.22% |
| GNN-MCTS-p | 3.92 | 0.09% | 5.77 | 1.47% | 8.06 | 4.24% | 3.31 | 0.14% | 3.44 | 1.70% | 3.54 | 4.39% |
| GNN-MCTS-v | 3.92 | 0.08% | 5.77 | 1.64% | 8.06 | 4.24% | 3.30 | 0.04% | 3.42 | 1.13% | 3.58 | 5.67% |
| GNN-MCTS$_{ave.}$ | 3.98 | 1.62% | 6.09 | 7.21% | 8.83 | 14.23% | 3.34 | 1.25% | 3.63 | 7.27% | 3.87 | 14.11% |

# 5 EXPERIMENTS

## 5.1 EXPERIMENTAL SETUP

### 5.1.1 BASELINES

To compute optimal solutions for both TSP, we use two state-of-the-art solvers, Concorde [4] (Applegate et al., 2006a) and Gurobi [5] (Optimization, 2013). We compare against Nearest, Random and Farthest Insertion, as well as Nearest Neighbor, which are non-learned baseline algorithms that also derive a tour by adding vertices successively. Additionally, we compare against excellent deep learning-based methods based on the greedy framework as mentioned in Section 2, most importantly Vinyals *et al.* (Vinyals et al., 2015), Bello *et al.* (Bello et al., 2017), Kool *et al.* (Kool et al., 2019), and Dai *et al.* (Dai et al., 2017).

### 5.1.2 TRAINING AND TESTING

We generate 50,000 instances (see in appendix A) for TSP20, TSP50, and TSP100, respectively, to train GNN (settings are in appendix B). We use state-of-art solvers (Gurobi and Concorde) to obtain the optimal tour sequence for each instance. Then we generate $N$ samples for each instance according to the optimal tour sequence. We divide the dataset into a training set, a validation set, and a test set according to the ratio of 8: 1: 1. We use Adam (Kingma & Ba, 2014) with 128 mini-batches and learning rate $10^{-3}$. Training proceeds for 30 epochs on a machine with 2080ti GPU. After training models for TSP20, TSP50, and TSP100, respectively, we use pre-trained GNN to guide GNN-MCTS. During testing, we randomly generate 1000 instances for the above three problems. The parameter settings of the GNN-MCTS used in our experiments are as follows: we set $c_{puct} = 1.3$ and $beam\ width = 1$ for three problems; we set $rollouts = 800, 800$ and $1200$ respectively for TSP20, TSP50, and TSP100.

## 5.2 RESULTS

Besides non-learned algorithms, we mainly compare our method with excellent deep learning-based works that derive tours on the greedy mechanism. We implement and train a Pointer network with supervised learning, but we find that our supervised learning results are not as good as those reported by (Vinyals et al., 2015). Results of Pointer network on the random instances are from the *optimality gaps* they report on 20, 50 vertex graphs. For other deep learning-based methods, we use experimental settings suggested by authors to train and get the same performance as reported.

---

[4]http://www.math.uwaterloo.ca/tsp/concorde/

[5]http://www.gurobi.com/

Rather than reporting the approximation ratio $\frac{c}{c^*}$, where $c$ is the objective solution value of tour S and $c^*$ is the best-known solution value of instance $G$, we use the average optimality gap $\frac{c-c^*}{c^*} = \frac{c}{c^*} - 1$ mentioned in (Kool et al., 2019). Table 1 reports the gap between the solution of each approach and the best-known solution for TSP20, TSP50, and TSP100. Our approach performs favorably against other methods up to 100 nodes on the "random" and "clustered" instances. Table 7 in appendix C reports the confidence interval on different confidence levels.

## 5.3 RUNNING TIMES

Running times are important but hard to compare because they can vary by two orders of magnitude as a result of implementation (Python or C++) and hardware (CPU or GPU). Our method is slower than other learning-based methods due to the look-ahead search. Our code is written by Python and we note that the MCTS procedure can speed up by rewritten code to C++. We test our algorithm, Gurobi and learning-based methods on a machine with 32 virtual CPU system (2 * Xeon(R) E5-2620)) and 8 * 2080ti. At each epoch, we test 32 instances in parallel and after 10 epochs, we report the time it takes to solve on each test instance (in Table 2).

## 5.4 GENERALIZATION TO LARGER PROBLEMS

In order to explore the generalization of our method, we train the GNN on TSP100 random instances and test our method on random instances including TSP200, TSP300 and TSP500. We mainly compare the learning-based methods proposed by Kool *et al.* and Dai *et al.* which made the best performance before our work respectively in Encoder-Decoder and Graph Embedding framework. The results (in Table 3) show that our algorithm could generalize to larger problems well than other learning-based algorithms even if trained in the small-scale instances.

## 5.5 ABLATION STUDY

### 5.5.1 AVERAGE VS BEST

We analyze the effect of different strategies used in the GNN-MCTS procedure. The comparison of different strategies are 1) **best.** Different from AlphaGo, we track the best action value found under each node's subtree for determining its exploitation value. At the end of several rollouts, we select the node with the best (biggest) action value as the next move in the root position. 2) **average.** As with the strategy used in AlphaGo, which is common in a two-player game, we track the average action value found under each node's subtree as exploitation value. Rather than selecting the node with the best (biggest) action value, we select the most visited node as the next move in the root position.

Table 1 shows the gap between solutions of our approach with two strategies and the best-known solution for TSP20, TSP50, and TSP100. We refer GNN-MCTS to denote "best" strategy and refer GNN-MCTS$_{ave.}$ to denote "average" strategy. The empirical results show that using the "best" strategy is far better than using the "average" strategy for TSP.

### 5.5.2 COMPONENT CONTRIBUTION ANALYSIS

We conduct a controlled experiment on the TSP test set to analyze how each component contributes to the presented approach. First, we use our GNN to generate solutions in a greedy way, i.e., selecting the vertex with the biggest prior probability at each step; we refer to this version as GNN-MCTS-t. Then we use a GNN-MCTS which replaces the value function $h(s)$ (see in algorithm

Table 2: Running times of different method

|  | TSP20 | TSP50 | TSP100 |
|---|---|---|---|
| Dai et al. | 0.007s | 0.018s | 0.043s |
| Kool et al. | 0.036s | 0.054s | 0.084s |
| Gurobi | 0.017s | 0.2s | 1.9s |
| Our | 3.2s | 6.6s | 31.4s |

Table 3: Gap of different methods on larger instances

|  | TSP200 | TSP300 | TSP500 |
|---|---|---|---|
| Kool *et al.* | 8.19% | 12.32% | 20.40% |
| Dai *et al.* | 11.11% | 11.70% | 11.84% |
| Our | **1.91%** | **2.99%** | **4.37%** |

Table 4: Gap of different beam width

|  | w=1 | w=5 | w=10 |
|---|---|---|---|
| TSP20 | 2.25% | 1.50% | 1.50% |
| TSP50 | 5.32% | 3.64% | 3.38% |
| TSP100 | 11.37% | 8.11% | 7.48% |

Table 5: Time cost of different beam width

|  | w=1 | w=5 | w=10 |
|---|---|---|---|
| TSP20 | 55ms | 265ms | 534ms |
| TSP50 | 147ms | 730ms | 1461ms |
| TSP100 | 323ms | 1639ms | 3338ms |

1) with random rollout to generate tours; we refer to this version as GNN-MCTS-v. Furthermore, we take the GNN prior out of the picture and initialize prior probability to 1 for newly expanded nodes; we refer to this version as GNN-MCTS-p. Lastly, a pure MCTS which removes GNN prior and value function is listed for comparison; we refer this version as MCTS.

Table 1 shows the gap between the solution of each approach and the best-known solution on different TSP problems. The results from GNN-MCTS-p and GNN-MCTS show that GNN prior could help MCTS to effectively reduce the search space so that MCTS can allocate more computing resources to nodes with high value. Furthermore, the results from GNN-MCTS-v and GNN-MCTS show that value function $h(s)$ can estimate the path length from the leaf node well and the MCTS, which uses a suitable value function, can perform better than using a random rollout. Lastly, the gap of performance between GNN-MCTS-t and GNN-MCTS shows that the developed MCTS can efficiently avoid algorithm falling into local optimal and plays an important role in enhancing the performance of our method.

### 5.5.3 ANALYSIS OF VALUE FUNCTION

We conduct experiments to explore the effects of different widths on the performance of value function. Since the beam width mainly affects the accuracy of value function, we use the result of value function as a measure and report the Gap as defined in Table 1. Specifically, we set beam width to 1, 5, 10 and test performance of the value function on random instances including TSP20, TSP50, and TSP100. We also count the time cost of the different settings of the beam width.

The experimental results of Table 4 and Table 5 show that as the beam width increases, the performance of the value function will get better while the time cost will become larger. We need to make a trade-off between accuracy and time cost.

### 5.6 COMPARISON WITH OTHER DEEP NEURAL NETWORKS

Compared with the basic GNN, our GNN integrates edge information for computing new node feature, and it should extract more information and perform well than basic GNN. To support this statement, we compare the performance of basic GNN and our GNN on random instances, including TSP20, TSP50, and TSP100. We generate tour sequences by using the neural network in a greedy way, i.e., selecting vertex with the biggest prior probability at each step. The performance of two GNN is reported in Table 8 (see in appendix D).

We also compare the performance of GEN and our GNN to support the key improvements made by our GNN. Similar to the above comparison experiment, we generate tour sequences by using neural network in a greedy way. The performance of GEN and our GNN is reported in Table 8. Furthermore, we use GEN and our GNN to guide MCTS separately on "random" and "clustered" instances, including TSP20, TSP50 and TSP100. We refer to MCTS with GEN as GEN-MCTS. Table 1 reports the quality of solutions and shows that MCTS can get shorter tours when guided by our GNN.

## 6 CONCLUSION

We proposed a graph neural network assisted Monte Carlo Tree Search (GNN-MCTS) for the classical traveling salesman problem. The core idea of our approach lies in converting the TSP into a tree search problem. To capture the local and global graph structure, we train a graph neural network (GNN) which integrates node feature and edge weight into the feature update process. Instead of using the prior probability output by GNN in a greedy way, we designed a GNN-MCTS to provide scouting simulation so that the algorithm could avoid being stuck into the local optimum. The exper-

imental results show that the proposed approach can obtain shorter tours than other learning-based methods. We see the presented work as a step towards a new family of solvers for NP-hard problems that leverage both deep learning and classic heuristics. We will release code to support future progress in this direction.

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

## A    INSTANCE GENERATION

To evaluate our method against other approximation algorithms and deep learning-based approaches, we use an instance generator from the DIMACS TSP Challenge (Johnson et al., 2001) to generate two types of Euclidean instances: "random" instances consist of $n$ points scattered uniformly at random in the $[10^6, 10^6]$ square; "clustered" instances include $n$ points that are clustered into $n/100$ clusters. We consider three benchmark tasks, Euclidean TSP20, 50 and 100, for which we generate a train set of 50,000 instances and a test set of 1,000 instances.

## B    GRAPH NEURAL NETWORK SETTINGS

Our GNN has $T = 3$ node-update layers, which is deep enough for nodes to aggregate information associated with their neighbor vertices. Since the input is a "tagged" graph with 9-dimensional feature on vertices, the input contains vectors of size $H^0 = 9$. The width of other layers are identical: $H^t = 64$ for $t = 1, 2$.

The proposed GNN has a deep architecture that consists of several node-update layers. Therefore, as the model gets deeper with more layers, the more information can be aggregated by nodes. We train proposed GNN with the different number of layers on random instance from TSP20. We greedily use the prior probability, i.e., selecting the vertex with the biggest prior probability, to derive tour sequence. We report Gap as defined in Table 1. The result of Table 6 show that the performance of GNN will become better as the number of network layers increases.

Table 6: Effect of the number of layers on random instances

| Layers | T=1 | T=2 | T=3 |
|---|---|---|---|
| Gap | 5.1% | 3.2% | 2.2% |

## C    CONFIDENCE INTERVAL ON DIFFERENT CONFIDENCE LEVELS

Table 7: Confidence interval on different confidence levels

| Confidence level | | 90% | 95% | 99% |
|---|---|---|---|---|
| Random | n=20 | 3.922±0.046 | 3.922±0.054 | 3.922±0.071 |
| | n=50 | 5.701±0.041 | 5.701±0.048 | 5.701±0.064 |
| | n=100 | 7.850±0.039 | 7.850±0.047 | 7.850±0.062 |
| Clustered | n=20 | 3.305±0.078 | 3.305±0.093 | 3.305±0.122 |
| | n=50 | 3.391±0.051 | 3.391±0.061 | 3.391±0.080 |
| | n=100 | 3.442±0.036 | 3.442±0.043 | 3.442±0.056 |

## D    PERFORMANCE OF DIFFERENT NETWORKS

Table 8: Performance of different networks

| | n=20 | n=50 | n=100 |
|---|---|---|---|
| GNN | 4.8% | 21.1% | 58.9% |
| GEN | 4.1% | 12.5% | 21.9% |
| Our | **2.2%** | **5.3%** | **11.3%** |

