# OpenReview forum: "A Graph Neural Network Assisted Monte Carlo Tree Search Approach to Traveling Salesman Problem"
_ICLR.cc/2020/Conference — Reject_

### Official Review · AnonReviewer3 · 2019-10-22
**Official Blind Review #3**

**Rating:** 6

**Review:**

In this paper, the authors introduce a new Monte Carlo Tree Search-based (MCTS) algorithm for computing approximate solutions to the Traveling Salesman Problem (TSP). Yet since the TSP is NP-complete, a learned heuristic is used to guide the search process. For this learned heuristic, the authors propose a Graph Neural Network-derived approach, in which an additional term is added to the network definition that explicitly adds the metric distance between neighboring nodes during each iteration. They perform favorably compared to other TSP approaches, demonstrating improved performance on relatively small TSP problems and quite well on larger problems out of reach for other deep learning strategies.

I believe that the paper is built around some good ideas that tackle an interesting problem; the Traveling Salesman Problem and variants are popular and having learning-based approaches to replace heuristics is important. In particular, choosing to use an MCTS to tackle this problem feels like a natural approach, and using a GNN as a learning backend feels like a encourage better performance with fewer training samples. However, there are too many questions raised by decisions the authors have made to warrant acceptance in the current state; I would be willing to revise my score if some more detailed analysis of these points were included.

First, the heuristic value function: this value function h(s) is defined in the appendix but should be motivated and described (in detail) in the text body. As written, this information is not included in the main body of the paper yet is critical for the implementation. Also, though it is intuitively clear why a random policy is unlikely to result in a poor result, it is never compared against; how does the performance degrade if the heuristic value function is not used? Finally, the parameter 'beam width' used in the evaluation of the value function but is only set to 1 in all experiments. Some experiments should be included to show how increasing beam width impacts performance (or the authors should provide a reason these experiments were not run). Finally, it seems as if there already exists heuristic methods (against which the paper compares performance); could these be used instead of this value function?

Additionally, how is the set of Neighbors defined? It is suggested in the text that it is not all nodes, but not using all nodes is a limiting assumption. Relatedly, it would be helpful if the authors could better motivate their additional term in Eq. (2); at the moment, though using the euclidian distance to weight the edges, it is unclear why this function is a better choice than something else, for instance a Gaussian kernel or a kernel with finite support. In addition, the authors motivate that the distance between nodes is very important for the performance of the system, yet the coordinates of each vertex are included as part of the input vector so that (in principle) the network could learn to use this information. A comparison against a network implemented using the basic GNN model, defined in Eq. (1), should be included to compare performance.

In summary, there are a few choices that would need to be better justified for me to really support acceptance. However, there are some quite interesting ideas underpinning this paper, and I hope to see it published.

Minor comments:
- Overall, I like the structure of the paper. At the beginning of all major sections there is an overview of what the remainder of the section will contain. This helps readability. I also like the comparison between the proposed work and AlphaGo, which popularized using deep learning in combination with MCTS; this enhances the clarity of the paper.
- The related work section would be more instructive if it also gave some information about the limitations of the alternative deep learning approaches and how the proposed technique overcomes these. My assumption is that all approaches discussed in the second paragraph are "greedy" and suffer from the limitations mentioned in the introduction. However, I am not sufficiently familiar with the literature to be certain. A sentence or two mentioning this or relating that work to the proposed MCTS approach would be informative.
- The last paragraph of the Related Work section, discussing the work of Nowak et al 2017 and Dai et al 2017, introduces some numbers with no context: e.g., "optimality gap of 2.7%". It is unclear at this stage if this number is good or bad. Some more context and discussion of this work might be helpful for clarity, particularly since the Nowak work seems to be the only other technique using GNN.
- Some general proofreading for language should be performed, as there are occasionally typos or missing words throughout the paper. Some examples: "compute the prior probability that indicates how likely each vertex [being->is] in the tour sequence"; "Similar to the [implement->implementation], in Silver..."; "[Rondom->Random]" in tables.
- In Sec. 4.1, it is unclear what is meant by "improved probability \hat{P} of selecting the next vertex".
- I believe there is an inconsistency in the description of the MCTS strategy. Though the action value is set to the 'max' during the Back-Propagation Strategy, the value of Q is initialized to infinity.

Suggestions for improvement (no impact on review):
- Clarity: the language in the 3rd and 4th paragraphs of the introduction [begins with "In this paper, ..."] could be made clearer.
  - The language "part of the tour sequence" is not quite clear, since, when the process is complete, all points will be in the tour. It should be made clearer that the algorithm is referring to a "partial tour" as opposed to the final tour. This clarity issue also appears later in Sec. 4.
  - "Similar to above-learned heuristic approaches..." It might be clearer if you began the sentence with "Yet," or "However," so that it is more obvious to the reader that you intend to introduce a solution to this problem.
- Equation formatting: Please use '\left(' and '\right)' for putting parenthesis around taller symbols, like \sum.
- When describing the MCTS procedure, I have seen the word "rollouts" used much more frequently than "playouts". Consider changing this language (though the meaning is clear).

**Experience Assessment:**

I have read many papers in this area.

**Review Assessment: Checking Correctness Of Derivations And Theory:**

I assessed the sensibility of the derivations and theory.

**Review Assessment: Checking Correctness Of Experiments:**

I assessed the sensibility of the experiments.

**Review Assessment: Thoroughness In Paper Reading:**

I read the paper thoroughly.

---

> ### Author Response · Authors · 2019-11-13
> **Response to Review #3 Part 1**
>
> Thank you for your constructive and encouraging comments. we address your concerns below:
>
> Question 1: “First, the heuristic value function: this value function h(s) is defined in the appendix but should be motivated and described (in detail) in the text body.”
> Answer 1: We accept your suggestion, and adjust the value function’s position to the corresponding place in the article's body in the new version.
>
> Question 2: “Also, though it is intuitively clear why a random policy is unlikely to result in a poor result, it is never compared against; how does the performance degrade if the heuristic value function is not used?”
> Answer 2: The results were included in Table 5, where SE-GNN+Tree_v denotes using the policy random and SE-GNN+Tree denotes using the value function. The description for Table 5 was not clear in the manuscript. We will revise the related part accordingly.
>
> Question 3: “Finally, the parameter 'beam width' used in the evaluation of the value function but is only set to 1 in all experiments. Some experiments should be included to show how increasing beam width impacts performance (or the authors should provide a reason these experiments were not run).”
> Answer 3: We conduct experiments to explore the effects of different widths on the performance of the algorithm. Since the beam width mainly affects the accuracy of the value function, we use the result of the value function as a measure and report the Gap as defined in Table 1. Specifically, we set beam width to 1, 5, 10, 20 and test performance of the value function on random instances including TSP20, TSP50, and TSP100. The experimental results are as follows: For TSP20, the Gap is 2.25%(1), 1.50%(5), 1.50%(10), 1.50%(20) ; For TSP50, the Gap is 5.32%(1), 3.64%(5), 3.38%(10), 3.22%(20); For TSP100, the Gap is 11.37%(1), 8.11%(5), 7.48%(10), 6.87%(20). We also count the time cost of the different settings of the beam width. The result of the time cost are as follows: For TSP20, 55ms(1), 265ms(5), 534ms(10), 1063ms(20); For TSP50, 147ms(1), 730ms(5), 1461ms(10), 2957ms(20); For TSP100, 323ms(1), 1639ms(5), 3338ms(10), 6820ms(20). The experimental results show that as the beam width increases, the performance of the value function will get better while the time cost will become larger. We need to make a trade-off between accuracy and time cost.
>
> Question 4:  Finally, it seems as if there already exists heuristic methods (against which the paper compares performance); could these be used instead of this value function?
> Answer 4: We conduct experiments about replacing value function with different heuristic methods including nearest insertion, farthest insertion and random insertion. We report the Gap as defined in Table 1. The results are as follows: For nearest insertion, TSP20(4.53%), TSP50(14.95%), TSP100(21.79%); For farthest insertion, TSP20(4.40%), TSP50(14.52%), TSP100(21.76%); For random insertion, TSP20(4.99%), TSP50(13.95%), TSP100(22.03%). The results show that the heuristic methods mentioned in the article are not suitable for our algorithm. We think that the partial tour corresponding to the leaf node in the tree suffers the performance of the above heuristic methods. Designing an effective evaluation function is indeed a very important direction for further research.
>
> Question 5: “How is the set of Neighbors defined?”
> Answer 5: Complete graph is constructed for TSP, so the set of neighbors of one node contains all nodes except itself. We will describe it with more details in the new version.
>
> Question 6: “Relatedly, it would be helpful if the authors could better motivate their additional term in Eq. (2);” “A comparison against a network implemented using the basic GNN model, defined in Eq. (1), should be included to compare performance.”
> Answer 6: We agree that in principle the neural network should have learned the distance information from the coordinates of the nodes. It is a simple thing for people, but our empirical results indicate that it is difficult for the neural network to learn the distance information.
> In the paper, we compare the basic GNN (no edge information) with SE-GNN in the following ways. Firstly, we compare the accuracy of two models on test data when training the neural network. And then we use the greedy policy that selecting node with biggest prior output by the network as next move to derive the tour. Table 6 reports the corresponding results and shows that adding distance information to the GNN can improve the performance of the model.
>
> Question 7: Alternative way to measure the distance between nodes.
> Answers 7: Like Gaussian kernel function, we use $e_{v,u}W^{t}_{3}$ to map Euclidian distance to high dimensional in Eq.(2).

---

> ### Author Response · Authors · 2019-11-13
> **Response to Review #3 Part 2**
>
> Question 8: “ The related work section would be more instructive if it also gave some information about the limitations of the alternative deep learning approaches and how the proposed technique overcomes these.”
> Answer 8: Thank you for the suggestion. We reorganized the related work. We agree with you that all the approaches discussed in the second paragraph are "greedy" and suffer from the limitations mentioned in the introduction. What’s more, we have made more context and discussion of Nowak et al 2017 and Dai et al 2017 and you will see that in the new version.
>
> Question 9: typos
> Answer 9: We will correct typos in the new version.
>
> Question 10: The meaning of the "improved probability \hat{P} of selecting the next vertex".
> Answer 10: It should be that “based on the improved probability \hat{P} generated by the GNN-MCTS”.
>
> Question 11: The value of Q is initialized to infinity.
> Answer 11: We are so sorry for our description to confuse you. The Q value only needs to be initialized to a small value, i.e., - infinity. In our code, we initialize Q value to -5.0 (TSP20), -10.0 (TSP50) and -15.0 (TSP100).
>
> Question 12: Suggestions for improvement
> Answer 12: We are so grateful for your suggestions and we will adjust the corresponding part in the new version.

---

### Official Review · AnonReviewer2 · 2019-10-24
**Official Blind Review #2**

**Rating:** 1

**Review:**

The paper proposes learning a TSP solver that incrementally constructs a tour by adding one city at a time to it using a graph neural network and MCTS. The problem is posed as a reinforcement learning problem, and the graph neural network parameters are trained to minimize the tour length on a training set of TSP instances. A graph neural network architecture called Static Edge Graph Neural Networks is introduced which takes into account the graph of all cities in a given problem instance as well as the partial tour constructed so far in an episode. The network predicts probabilities for the remaining cities to be selected as the next city in the tour, which is then used to compute a value function that guides MCTS. Results on synthetic TSP instances with 20, 50, and 100 cities show that the approach is able to achieve better objective values than prior learning-based approaches. Applying AlphaZero-like approaches to TSP is an interesting test case for understanding how well they can work on hard optimization problems.


The paper has several drawbacks:
- The evaluation seems to be flawed as there is no mention of running time of the various algorithms being compared anywhere in the text. It’s not possible to make a fair comparison without controlling for running time. As an extreme example, even random search will eventually find the global optimum if given sufficient time. So the results are not very meaningful without the running times.

- Novelty is fairly low. The changes in SEGNN compared to previous works are incremental or not novel, and the overall idea is the same as AlphaGo/Zero. While I don’t think novelty is a strict requirement, if it is absent, then it should be compensated with strong empirical results, but the paper lacks that as well.

- A discussion on whether the approach can plausibly scale to much larger TSP instances is missing. First, there is the question of whether learning can succeed on much larger instances. Second, even if good policies can indeed be learned, can they provide competitive running times compared to the state-of-the-art TSP solvers? Graph net inference’s compute cost scales linearly with graph size (number of cities), and since multiple inference passes need to be performed per step (to pick the next city to add to the current partial tour), the overall cost scales quadratically. This is worse than the empirical scaling of solvers like LKH and POPMUSIC. One has to consider approaches with cost that scales roughly linearly to be able to compete with state-of-the-art solvers. It should be noted that TSP instances with <= 100 cities are really trivial for the best solvers, and outperforming them with a learning-based approach may not be plausible until much larger instances are considered (e.g., > 10K cities). The ML community needs to move away from evaluating on small instances if the long term goal is to beat state-of-the-art solvers with learning.


Additional comments:
- There are a lot of typos. A few that I caught: Tables 1 and 7 say “Rondom”, “approximation ration”, “ReLu”, “provides a heuristics”, “Similar to the implement”.

- Table 6 gives the highest test accuracy during training, but this could be misleading (e.g., there could be random spikes in test performance during training). A smoother metric should be used.

- Table 3 title is confusing.


**Experience Assessment:**

I have read many papers in this area.

**Review Assessment: Checking Correctness Of Derivations And Theory:**

I assessed the sensibility of the derivations and theory.

**Review Assessment: Checking Correctness Of Experiments:**

I assessed the sensibility of the experiments.

**Review Assessment: Thoroughness In Paper Reading:**

I read the paper at least twice and used my best judgement in assessing the paper.

---

> ### Author Response · Authors · 2019-11-15
> **Response to Review #2**
>
> Thank you for the detailed review and helpful suggestions. We address your concerns below:
>
> Question 1: Running time of the algorithm.
> Answer 1: Thank you for this suggestion (which was also noted by reviewer 4), this is indeed something that was missing which we have supplemented the running time of our algorithm, Gurobi, and other learning-based methods. Running times are important but hard to compare: they can vary by two orders of magnitude as a result of implementation (Python vs C++) and hardware (CPU vs GPU). Our algorithm is slower than other learning-based algorithms due to the look-ahead search. Our code is written by Python and we note that the MCTS procedure can speed up by rewritten code to C++. We test our algorithm, Gurobi and learning-based methods on a machine with 32 virtual CPU systems (2 * Xeon(R) E5-2620)) and 8 * 2080ti. At each epoch, we test 32 instances in parallel and after 4 epochs, we report the time it takes to solve on each test instance. The results are as follows:
> 				TSP20		TSP50		TSP100
> - Our  			3.2s  		6.6s			31.4s
> - Gurobi			0.017s   	        0.2s			1.9s
> - Dai et al		0.007s		0.018s		0.043s
> - Kool et al		0.036s		0.054s		0.084s
>
> Although the cost time of our algorithm is not as fast as the traditional optimizer such as Gurobi, our algorithm has a good generalization ability than other learning-based algorithms.
>
> Question 2: “The discussion on whether the approach can plausibly scale to much larger TSP instances”.
> Answer 2: We conduct the experiment to explore the performance of our algorithm and other learning-based algorithms on larger problems. We made some changes to make our algorithm work on larger instances.
>   Firstly, we revise the Eq.(4) to $f(G|S;\Theta)=\text{softmax}(sum(H_{1}^{T}),…,sum(H_{n}^{T}))$, where sum denotes summation operator. This change allows our network to inference on large-scale instances after training on small-scale instances.
>   Secondly, in order to reduce the amount of memory used in the MCTS procedure, we only expand the top ten child nodes based on the prior probability output by SE-GNN.
>   We retrain our SE-GNN on TSP100 random instances and test our algorithm using the pre-trained model (TSP100) on random instances including TSP200, TSP300, and TSP500. We mainly compare the learning-based algorithm proposed by Kool et al. and Dai et al. which made the best performance before our work respectively in Encoder-Decoder and Graph Embedding framework. We report the Gap as defined in Table 1. The results are as follows:
>
>                       TSP200        TSP300	 TSP500
> Our               1.91%           2.99%		  4.37%
> Kool et al.    8.19%           12.32%          20.40%
> Dai et al.      11.11%	    11.70%         11.84%
>
> The results show that our algorithm could generalize to larger problems well than other learning-based algorithms even if trained in the small-scale instances. Scalability is indeed a very important direction for further research. We think that the way that heuristics (like you mention) scale almost linearly is by considering the problem locally, e.g. by local search or by limiting the set of edges for nodes (e.g. consider a sparse graph). Unlikely the previous work that directly using the deep neural network to construct a tour, we combine the neural network with the classic local search method (MCTS). We see the presented work as a step towards a new family of solvers for NP-hard problems that leverage both deep learning and classic heuristics.
>
> Question 2: Adding more empirical results.
> Answer 2: Based on the comments of reviewer 3 and reviewer 4, we added more experiments about GNN and GNN-MCTS in the new version.
>
> Question 2: typos
> Answer 2: We will correct typos in the new version.
>
> Question 3: Misleading metric in Table 6.
> Answer 3: We agree with you and have removed Acc* metric.
>
> Question 4: “Table 3 title is confusing”.
> Answer 4: We change the title to ‘’Confidence interval on different confidence levels

---

### Official Review · AnonReviewer4 · 2019-11-01
**Official Blind Review #4**

**Rating:** 6

**Review:**

EDIT: After the authors response and update of the contributions to indicate that the main contribution of the paper is the application of GNNs and MCTS to the TSP (rather than the original claims that that the model architecture and search approach were novel contributions), I increased my score from Weak Reject to Weak Accept. However, given that the paper is now more focused on solving the TSP, and I am not an expert on that specifically I had to reduce my experience assessment, as while I am more confident now that the paper is technically correct, it is harder for me to judge if the paper should be accepted in terms of empirical strength since I am not familiar with TSP baselines.

The authors propose an MCTS-based learned approach using Graph Neural Networks to solve the traveling salesman problem (TSP) agents.

The authors write the TSP as an MDP where the state consists of the nodes visited by the agent and the last node visited by the agent, the action consists of selecting the next node to visit, and the reward at each step is the negative cost of the travel between the last node and the next node.

The learned part of the model uses a “static-edge graph neural network” (SE-GNN). This network allows to access the full graph context, including edge features, to make node predictions. This is listed as the first paper contribution. At train time, this network is trained to predict the probability of each unvisited node to be next in the optimal path. This is trained via supervised learning using optimal paths precomputed with state of the art TSP solvers.

At test time, they use MCTS with a variant of PUCT, where the pre-trained SE-GNN is used as the prior policy, and there is a selection strategy during search that balances the prior probability, and the Q values estimated by MCTS, using max based updates (e.g. during back up new Q estimates replace old estimates if and only if the are larger than the previous ones). This is listed as the second paper contribution. Authors show that the approach beats other learned solvers in the TSP problem by a large margin in terms of optimality gap.

While I think the work is interesting, I am not sure that what the authors cite as main contributions of the paper are truly the main contributions. In my opinion the main contribution would be the state of the art performance at solving the TSP using learned methods. I cannot, however, recommend acceptance due to the following reasons.

With respect to the first claim “SE-GNN that has access to the full graph context and generates the prior probability of each vertex”, there are already many models that allow to condition on edge features, including InteractionNetworks, RelationNetworks and GraphNetworks. This paper has a good overview of this family of methods and most of them allow to access the full graph context too (https://arxiv.org/abs/1806.01261). Most of these models are very well known and are in principle more expressive than the one proposed in this paper, and allow generalization to different graph sizes, so the motivation for introducing a new model is not very clear, specially if these baselines are not compared.

With respect to the MCTS contribution at test time, it seems that the changes made to the algorithm compared to AlphaGo, are very specific to the TSP and there is not much discussion about which other sort of problems may benefit from the same modifications, so it is hard to evaluate its value as a standalone contribution independent from the TSP.

On the basis of state of the art performance at solving the TSP using learned methods:
* The model requires access to a dataset with optimal solutions to train it, and I doubt it can solve the problems faster than Gurobi in terms of wall time. For this result to be more interesting, the authors should be able to show that the model can generalize to larger problems (where the combinatorial complexity may start making approaches like Gurobi struggle). However it is not clear if the model can generalize to larger graphs.
* Beyond that I am not an expert on TSP specifically, and I don’t know the TSP literature, so I cannot give a strong recommendation.

There are some additional papers that may be relevant to this line of work:
* (MIP, NeurIPS 2019) Learning to branch in MIP problems using similar technique pretraining a GNN and use it to guide a solver at test time (no MCTS though) (https://arxiv.org/abs/1906.01629)
* (SAT, SAT Conference 2019) Learning to predict unsat cores (similar to the previous one but for SAT problems) (https://arxiv.org/abs/1903.04671)
* (Structural construction, ICML 2019) Building graphs by choosing actions over the edges of a graph solving the full RL problem end to end, and also integrating MCTS with learned prior both at train time and test time (together and independently) (http://proceedings.mlr.press/v97/bapst19a/bapst19a.pdf)

Some additional typos/ feedback:
* It would be good to have a pure MCTS baseline with not learned prior as an additional ablation (e.g. taking the SE-GNN prior out of the picture).
* In the “Selection Strategy” paragraph, the action is said to be picked as argmax(Q + U), where U is proportional to the prior for each action. However, Q is said to be initialized to infinite. This would mean that at the beginning of search all actions will be tied at infinite value, and my default assumption would be that in these conditions an action is chosen uniformly at random. I suspect what happens in this case is that the action with the highest prior is picked to break the tie at infinite, however if this is the case this should be indicated in the math.
* In the “Expansion Strategy” paragraph, the Q values are said to be initialized to infinite. However in the Back-Propagation strategy it is said they are updated using newQ = max(oldQ, value_rollout). If this was true the values would always remain infinite, I assume the max is not applied if the previous value was still infinite.
* In the “Play” paragraph: The action is said to be picked according to the biggest Q value at the root, I assume in cases where the planning budget is smaller than the number of nodes, and not all actions at the root are explored, the actions that have not been explored are masked out.
* Non-exhaustive list of typos: “Rondom” —> “Random”, “provides a heuristics” —> “provides a heuristic”, “strcuture2vec” — > “structure2vec”, weird line break at top of page 8.



**Experience Assessment:**

I have read many papers in this area.

**Review Assessment: Checking Correctness Of Derivations And Theory:**

I assessed the sensibility of the derivations and theory.

**Review Assessment: Checking Correctness Of Experiments:**

I assessed the sensibility of the experiments.

**Review Assessment: Thoroughness In Paper Reading:**

I read the paper at least twice and used my best judgement in assessing the paper.

---

> ### Author Response · Authors · 2019-11-15
> **Response to Review #4**
>
> Thank you for seeing the importance of the problem and the value of showing the broad applicability. Please let us address your concerns.
>
> Question 1: “The motivation for introducing a new model is not very clear, specially if these baselines are not compared.”
> Answer 1: In this paper, our intention is not to introduce a new GNN model and beats other well-known models, but to use GNN to extract features for TSP. Rather than using the basic GNN, we integrate edge information into the GNN and empirical results show that the incremental chance can improve the feature extraction ability of the GNN. We have modified the representation of the corresponding part in the article. In the feature, we will explore other GNN models as you mentioned to improve the feature extraction and generalization capabilities of our method.
>
> Question 2: Could other sorts of problems benefit from the GNN-MCTS?
> Answer 2: In this paper, we focus on the traveling salesman problem and we will extend the proposed MCTS to other combinatorial optimization problems in the future work.
>
> Question 3: Running time and generalization of the algorithm.
> Answer 3: Running time and generalization of the algorithm are also noted by Reviewer 2 and please see our response to Reviewer 2 on question 1 and question 2.
>
> Question 4: “It would be good to have a pure MCTS baseline with not learned prior as an additional ablation (e.g. taking the SE-GNN prior out of the picture).”
> Answer 4: Thanks for your proposal. We have added the pure MCTS baseline in the new version.
>
> Question 5: Question about Q value initialization.
> Answer 5: We are so sorry for our description to confuse you. The Q value only needs to be initialized to a small value, i.e., - infinity. In our code, we initialize Q value to -5.0 (TSP20), -10.0 (TSP50) and -15.0 (TSP100).
>
> Question 6: “planning budget is smaller than the number of nodes, and not all actions at the root are explored, the actions that have not been explored are masked out. ”
> Answer 6: In the MCST procedure, only nodes with high value (Q+U) will be explored multiple times. So, the MCTS can allocate more exploration resources to the direction of the possible optimal solution. By using PUCT, the small prior child nodes are rarely visited, and the solution space can cut down in this way. In the “play” phase, we pick action according to the biggest Q value at the root and mask out the actions that have not been explored because these nodes have very small prior or Q value.
>
> Question 7: typos
> Answer 7: We will correct typos in the new version.

---

> > ### Comment · AnonReviewer4 · 2019-11-15
> > **Response to authors**
> >
> > Thank you for addressing my comments, see some replies below.
> >
> > Q1: Thank you for clarifying and updating the contributions in the introduction. I think they now reflect better the contributions of the paper.
> >
> > Q3: Thank you for the additional experiments, the results on generalization specifically seem very valuable.
> >
> > Q5: Thank you for updating the description, and changing it from +inf to -inf, which is drastically different. However I still find it very confusing that on one hand you say you set it to -inf, and then on the footnote you say that you actually set it to -5, -10, and -15, depending on the problem. This sounds to meet that in that case the initial q value is just another hyparameter and it should be indicated as such. I wonder if you could just set it to a fixed value of 0, and then tune the c_puct constant separately. By the way, what is the value of c_puct used? I don't think I could find it on the paper, you may consider adding it.
> >
> > Q4: The main reason I asked abut Q5 was the confusion caused by the "bug" pointed in Q5. If the Q values were initiated at +Inf, this would mean the policy would favor exploration over exploration a lot, which is why I thought that the pure MCTS baseline could do better. Where the hyperparameters, and changed tuned for the pure MCTS baseline, though? I would expect tuning c_puct and tuning the initial q value would be very important here.
> >
> > Q6: Thanks, I was asking this because if the Q values had been initialized to "+Inf" as indicated in the previous version then all actions would be explored first, so I thought in your case you may have never been under the circumstances of having unexplored actions at the root.
> > From the reply: "In the “play” phase, we pick action according to the biggest Q value at the root and mask out the actions that have not been explored because these nodes have very small prior or Q value." What do you mean by "because these nodes have very small prior or Q value.", are you saying that they will be implicitly masked out because any action that is explored will already have higher value that the prior initial value, or do you explicitly mask out the actions. If it is the first then you should probably indicate in the paper that care should be taken when choosing the initial value depending on the reward structure so this still holds true. If it is the second, you should probably include a sentence saying that in the Play section.
> >
> > Finally, I would strongly recommend checking out and referencing some of the papers indicated on the review, particularly those on classic combinatorial problems (SAT and MIPs). Since this work is very similar in spirit to those but adding MCTS on top, and the approaches that they used on those could probably be augmented by MCTS too.

---

### Decision · Program_Chairs · 2019-12-19

**Decision:**

Reject

**Comment:**

The paper is a contribution to the recently emerging literature on learning
based approaches to combinatorial optimization.
The authors propose to pre-train a policy network to imitate SOTA solvers for
TSPs.
At test time, this policy is then improved, in an alpha-go like manner, with
MCTS, using beam-search rollouts to estimate bootstrap values.

The main concerns raised by the reviewers is lack of novelty (the proposed
algorithm is a straight forward application of graph NNs to MCTS) as well a the
experimental results.
Although comparing well to other learning based methods, the algorithm is far
away from the performance of SOTA solvers.

Although well written, the paper is below acceptance threshold.
The methodological novelty is low.
The reported results are an order of magnitude away from SOTA solvers, while previous work
has already reported the general feasibility of learned solvers to TPSs.
Furthermore, the overall contribution is somewhat unclear as the policy relies
on pre-training with solutions form existing solvers.